# Experimental and Numerical Investigations into Heat Transfer Using a Jet Cooler in High-Pressure Die Casting

Jan Bohacek [1,*], Krystof Mraz [1], Vladimir Krutis [2], Vaclav Kana [2], Alexander Vakhrushev [3], Ebrahim Karimi-Sibaki [3] and Abdellah Kharicha [3]

[1] Heat Transfer and Fluid Flow Laboratory, Faculty of Mechanical Engineering, Brno University of Technology, 61669 Brno, Czech Republic; krystof.mraz@vut.cz

[2] Department of Foundry Engineering, Faculty of Mechanical Engineering, Brno University of Technology, 61669 Brno, Czech Republic; vladimir.krutis@vut.cz (V.K.)

[3] Christian-Doppler Laboratory for Metallurgical Applications of Magnetohydrodynamics, Montanuniversität Leoben, 8700 Leoben, Austria; alexander.vakhrushev@unileoben.ac.at (A.V.); ebrahim.karimi-sibaki@unileoben.ac.at (E.K.-S.); abdellah.kharicha@unileoben.ac.at (A.K.)

*  Correspondence: jan.bohacek@vut.cz

**Abstract:** During high-pressure die casting, a significant amount of heat is dissipated via the liquid-cooled channels in the die. The jet cooler, also known as the die insert or bubbler, is one of the most commonly used cooling methods. Nowadays, foundries casting engineered products rely on numerical simulations using commercial software to determine cooling efficiency, which requires precise input data. However, the literature lacks sufficient investigations to describe the spatial distribution of the heat transfer coefficient in the jet cooler. In this study, we propose a solver using the open-source CFD package OpenFOAM and free library for nonlinear optimization NLopt for the inverse heat conduction problem that returns the desired distribution of the heat transfer coefficient. The experimental temperature measurements using multiple thermocouples are considered the input data. The robustness, efficiency, and accuracy of the model are rigorously tested and confirmed. Additionally, temperature measurements of the real jet cooler are presented.

**Keywords:** die casting; jet cooling; jet cooler; bubbler; bayonet; die insert; OpenFOAM; NLopt; IHCP; concave surface; spherical surface

## 1. Introduction

High-pressure die casting (HPDC) is a common process for casting nonferrous metals. It is particularly suited for the high-volume production of complex near-net shapes [1,2]. The majority of casting alloys are based on the secondary alloys Al-9Si and Al-12Si [3]. HPDC is characterized by two important features: (i) high turbulence experienced by the molten metal as it is fed at high speed into a die and (ii) a very rapid rate of solidification [4]. The first may result in entrapped air and the formation of gas porosity in castings. The second may appear as porosity due to metal shrinkage or cold shuts.

Gigacasting is currently probably the hottest trend in car manufacturing (Tesla, NIO, XPeng). It is basically the HPDC, but the machines are huge and work under enormous pressure. Hence, any earlier known issues inherent to the HPDC, even those insignificant ones, will soon ramp in importance and scale up.

Bigger foundries often rely on casting simulations. The more accurate the input for them, the more accurate the output, hence, the result of the simulation. The quality of the casting depends on the final solidified structure of the alloy. The solidified structure is determined by the temperature gradient and the isotherm velocity in the molten alloy. Therefore, the way the heat is extracted from the mold is crucial. Since the majority of the heat is dissipated via water cooling, extra care must be taken to accurately model the system of cooling. In a typical casting simulation, the flow of the coolant will not be

resolved. Instead, heat transfer coefficients (HTCs) will be imposed along with reference temperatures as thermal boundary conditions on relevant heat transfer surfaces. The field of HTC often varies spatially and temporally and is a function of the surface temperature. Hence, proper correlations and experimental or numerical approaches are vital to obtain accurate input data for numerical simulation.

The solidification of the metal is controlled using internal cooling passages. Nowadays, the so-called conformal cooling [5], fabricated at the same time as the die using the technology of additive manufacturing, is a promising direction for ultimate temperature control during the HPDC. Nevertheless, a typical die will still be classically drilled through to create channels through which the coolant will pass and extract the heat from the die. Jet coolers, also known as bubblers, are installed in blind holes that are perpendicular to the side of dies, with a tiny inner tube located in the center of each unit. Unlike the classically drilled cylindrical channels, jet coolers can dissipate heat more effectively. More importantly, they can be installed at specific, hardly accessible, locations in the die, requiring precise temperature control. Operating parameters such as the coolant flow rate, temperature, and opening and closing times can be adjusted as desired. Before being put in operation, the whole system must be optimized in order to produce defect-free products [6]. The casting simulations become very assistive at this stage. Most of the numerical models solve a system of PDEs requiring the appropriate boundary conditions (BCs). It is complex to define BCs for jet coolers and cooling channels, which are the main heat extractors in casting. Before casting, dies are typically preheated to a temperature around 200 °C, while the channels do not contain any coolant yet. The appearance of a boiling could be critical at the starting phase of the jet cooling. Therefore, an accurate and detailed analysis of the heat transfer in the HPDC process is mandatory at the design stage of the casting and specifically of the die system.

Single-phase convective heat transfer at the "tip" of the jet was investigated using CFD methods by Karkkainen and Nastac [7]. The numerical model predicts a large increase in heat transfer close to the bend in the jet impingement zone. In the annular region, yet quite far from the tip of the jet, the simulated heat transfer coefficient approaches the Gnielinski empirical correlation [8]. The heat transfer is, however, dampened at the stagnation point located at the hemispherical tip. The boiling phenomenon is not mentioned in the study [7].

Kawahara and Nishimura [9] predicted the convective HTC based on an analogy with the mass transfer coefficient obtained from the current passing through an electrolyte solution and multiple electrodes positioned around the hemispherical tip of the jet cooling pipe. In addition, the model was made of acrylic resin to facilitate the flow visualization of the working fluid with the aluminum powder. The trend of the local heat transfer coefficient agrees with that found in [7]. However, it is also limited to the flow of a single-phase liquid.

Sun and co-workers [10] have carried out experimental research to explore the heat transfer of bubblers and baffles. The insert was dipped into a molten aluminum bath (732 °C), while a single temperature was recorded (2 Hz) relatively far from the hemispherical tip at a distance of 21.5 mm. The results proved that the bubblers show better cooling efficiencies than the baffles. The cooling time correlation was suggested.

Fu [11] highlighted the thermal boundary condition of die inserts to be the essential parameter for casting simulations. Based on the heat transfer measurements of jets by Poole and Krane, the inverse task calculations were performed using a method from Taler and Zima [12]. Consequently, two correlations were obtained for the Nusselt number in the thesis [11,13], which can be used to determine the average heat transfer coefficient around the hemispherical tip and for a certain length of the annular section of the jet. The heat transfer coefficient of the hemispherical tip is significantly smaller by two orders of magnitude.

Unlike the present topic of jet coolers used in the HPDC, heat transfer from a jet impinging upon the concave cylindrical surface attracts more attention from researchers due to its very important application in the intensive cooling of the leading edge of a

turbine airfoil. Often, an array of circular jets [14] is substituted with a rectangular slot, which is favored by the geometry of the airfoil [15,16].

A typical jet cooler will always extract most of the heat at the jet impingement zone located around the symmetry axis of the semi-spherical tip for two reasons. First, it is positioned close to the contact with the hot casting, which provides a high-temperature difference and hence potentially high cooling intensity. Second, it is well known that the highest cooling intensity of an impinging jet is found in the stagnation point. Then, it is interesting to mention two studies [17,18], in which, respectively, a chevron jet and a lobed nozzle were used to increase the Nusselt number in the stagnation point. According to the authors, it makes sense to sacrifice a bit of the cooling intensity in the wall jet region for the good of cooling in the stagnation point, where higher temperature differences will further elevate the heat transfer. This could stand as a hint of how to improve the design of a jet cooler normally comprise a sharp-edged orifice of the inlet tube. Another inspiration could be found in [19], in which small dimples were used over the originally flat surface to promote heat transfer. Note that similar to the dimples, protrusions can also lead to the same effect of heat transfer enhancement [20]. A more complex, although presumably more efficient, cooling configuration was suggested in [21], namely a double-wall cooling structure with a reverse circular jet impingement. There yet exists an alternative to the above-mentioned shape (or geometrical) enhancements of the heat transfer of jet cooling. In [22], an intermittent turbulent impinging round jet was investigated experimentally to determine the influence of the pulse frequency and the pulse duration on the Nusselt number. It was found that the intermittent jet can lead to both outcomes: decreasing and increasing (50%) of the heat transfer.

A noteworthy remark on the comparison between a jet impingement upon a flat and concave surface can be found in the literature. One says the concave surface supports the thinning of the boundary layer and generation of the Taylor–Görtler vortices, resulting in better mixing and, thus, a better heat transfer [23]. Others argue that in the case of the concave surface, the flow recirculation causes the average temperature of the impinging jet to rise, therefore, lowering the thermal performance [24]. It is likely that both phenomena compete with each other.

With regard to jet cooling, none of the available studies seem to explain the local HTC behavior and give recommendations to assess proper thermal boundary conditions. Piecewise constant heat transfer coefficients are unlikely to help in assessing accurate casting simulations. The reference to the boiling phenomenon in jets is seldom found. Therefore, the objective of the present work is to obtain the spatial distribution of local HTCs for jet cooling by conducting temperature measurements and performing the subsequent inverse task calculations [25].

## 2. Materials and Methods

### 2.1. Temperature Measurements

The schematic drawing of the experimental apparatus is shown in Figure 1a. A photo of the apparatus is shown in Figure 1c.

The die insert (JW220 Jiffy-Tite cascade junction) is shown in the lower part of Figure 1a. The coolant enters through an inlet to a central mini-tube and impinges on the hemispherical tip of the insert. Then, the coolant makes a U-turn and continues flowing in the annular region to eventually leave through an outlet. The flow path around the hemispherical tip is shown as blue arrows in Figure 1a. In the same figure, the blind hole with the hemispherical tip (a red solid line) was machined by drilling and subsequent boring to achieve more precise dimensions and a better surface finish (Ra 3.2 μm).

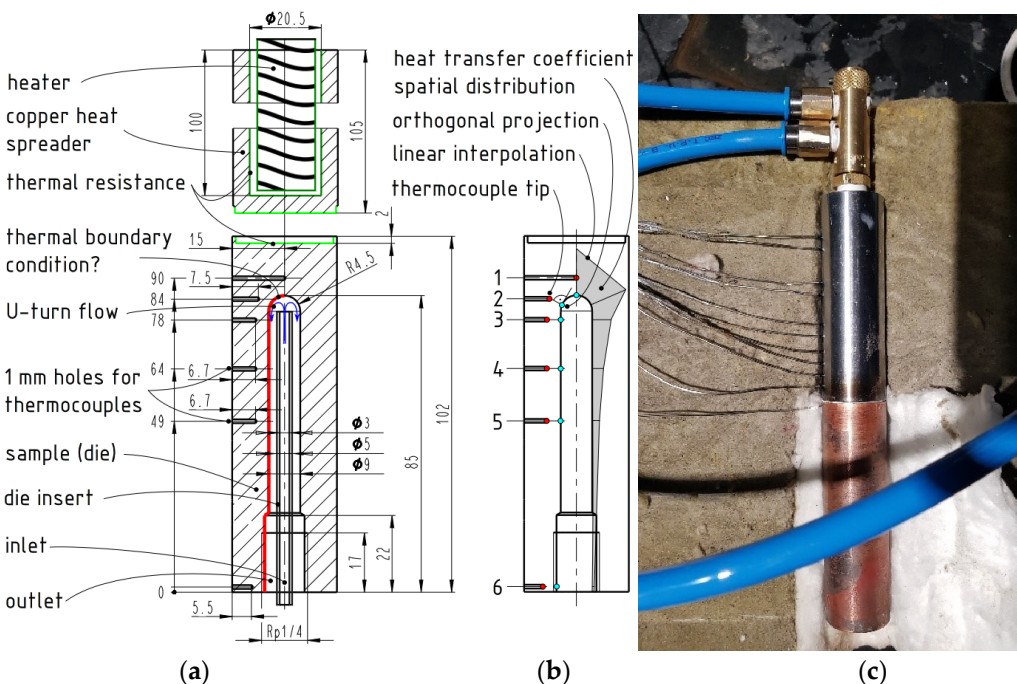

**Figure 1.** Die insert used in the study: (**a**) experimental configuration with dimensions; (**b**) schematics of the simulated inverse heat conduction problem in 2D with the thermocouples numbered from 1 to 6; (**c**) a photo of an experimental setup.

The sample was made of a hot work tool steel 1.2343 (Dievar). It was equipped with six K-type thermocouples (∅1 mm). In addition, two thermocouples were placed at the inlet and outlet to monitor the coolant temperature. The temperatures were recorded using ALMEMO 2890-9 datalogger with a sampling rate of 5 s. The flow rate of water was continuously measured with Krohne flow meter. A cylindrical electrical heater (1.6 kW, max. temperature 600 °C) was used as a heat source to emulate a hot aluminum melt sitting on the top of the sample, although knowing that the pouring temperature is somewhat higher, around 700 °C. A copper part with a blind hole for the heater is used to distribute the heat uniformly. The copper part with the heater inside is attached to the sample. A thin layer of heat-conductive paste is applied between the sample and the copper to ensure better thermal contact. The PID controller is used to keep the copper at a constant temperature, measured 1 mm from the contact surface (light green in Figure 1a). Due to the complicated regulation, the cooling power of the die insert is calculated on the water side. It is computed from the temperature difference at the inlet and outlet, the mass flow rate, and thermophysical properties of water evaluated at the average temperatures in the experiment.

A summary of Individual steps during the experiment can be written as follows:

1. The experiment starts together with the temperature recordings.
2. The water jet cooling is performed with a constant flow rate.
3. The heating from the top part of the sample is initiated.
4. The thermal steady state is achieved throughout the experiment.
5. Temperature cooling curves are saved for the subsequent inverse calculation and post-processing of the heat dissipation rate (also referred to as the cooling power).

The measured steady-state temperatures are shown along with other important data in Table 1. Steady-state readings from thermocouples are also plotted in Figure 2. In Table 1, the flow rates correspond to the common range of values used with the given die-insert JW220.

**Table 1.** Experimental settings and the measured steady-state temperatures.

|  |  | Exp. #1 | Exp. #2 | Exp. #3 | Exp. #4 |
|---|---|---|---|---|---|
| Flow rate | L/h | 90 | 120 | 230 | 300 |
| Inlet temperature | °C | 69.6 | 69.7 | 70.2 | 70.2 |
| Outlet temperature | °C | 70.8 | 70.5 | 71.0 | 70.7 |
| Cooling power | W | 120 | 120.0 | 213 | 190 |
| Copper temperature | °C | 600 | 600 | 550 | 550 |
| | | | | | |
| Steady-state readings | #1 | 173.8 | 171.1 | 229.0 | 211.2 |
| from thermocouples (°C): | #2 | 139.1 | 136.7 | 173.4 | 162.1 |
| | #3 | 107.2 | 105.3 | 127.7 | 120.9 |
| | #4 | 80.6 | 79.5 | 84.8 | 82.6 |
| | #5 | 73.1 | 72.5 | 73.8 | 72.9 |
| | #6 | 70.9 | 70.5 | 70.5 | 70.1 |

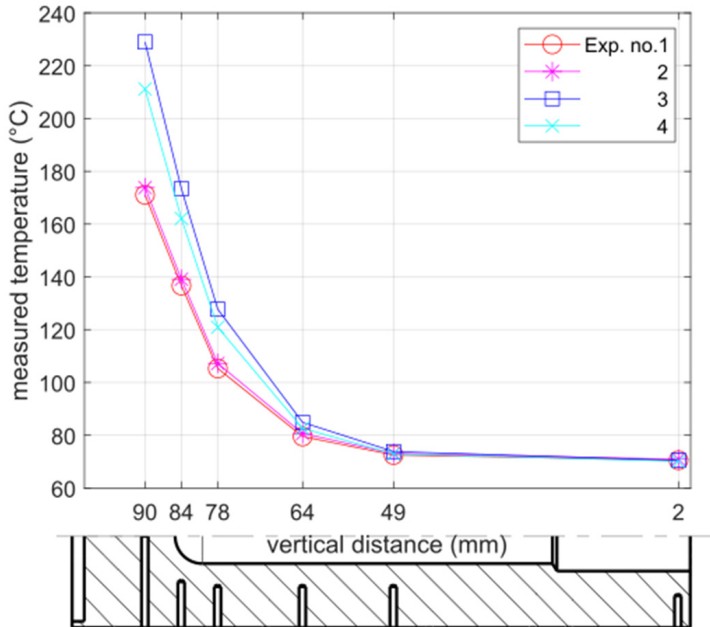

**Figure 2.** The steady-state temperatures recorded with the thermocouples #1–6 in the experiments Exp. #1–4. Vertical distance is aligned with geometry.

During the first two experiments, Exp. #1 and Exp. #2, the controller was set to maintain the copper temperature at 600 °C, which ultimately led to permanent damage to the heater. The heater was replaced with a new one, and the whole experimental device was reassembled. Due to this reassembling step, the thermal resistances changed (shown as green lines in Figure 1a). Moreover, during the last two experiments, Exp. #3 and Exp. #4, the copper temperature was decreased to 550 °C to avoid burning the heater. For this reason, one should not look for a correlation between Exp. #1–2 and #3–4. This fact, however, by no means inhibits subsequent inverse calculations and eventual obtaining of the meaningful heat transfer coefficients at the surface, shown in red in Figure 1a.

### 2.2. Inverse Heat Conduction Problem in 2D

In Bohacek [26], a sequential inverse heat conduction problem (IHCP) solver was developed in the open-source CFD code OpenFOAM [27] using the external optimization library NLopt [28].

A spatially uniform transient heat transfer coefficient of a nozzle spray was reconstructed using the temperature record from a single thermocouple. The present IHCP model is built upon this solver. The major differences are as follows: (i) the steady-state

heat conduction equation is considered, and (ii) records from multiple thermocouples (shown as red markers in Figure 1b) are used to obtain spatial distribution of the heat transfer coefficient on the surface of interest (highlighted in red in Figure 1a).

As the present IHCP solver changed only a little compared to the one introduced in [26], herein, only the fundamental changes are highlighted and discussed. The governing equation is the stationary heat conduction equation solved in the space domain $\Omega$ and can be written as follows:

$$\nabla\cdot(k\nabla T) = 0 \qquad (\in \Omega), \tag{1}$$

in which symbols $T$, $k$, $\nabla$, and $\nabla\cdot$ denote, respectively the temperature, the thermal conductivity, the gradient, and the divergence operator. The axisymmetric computational geometry with the grid is shown in Figure 3a. In the same figure, the thermal boundary conditions (BC) are highlighted in color: the adiabatic BC in black $\left(-k\frac{\partial T}{\partial n} = 0\right)$, the heat flux BC in green $\left(-k\frac{\partial T}{\partial n} = q\right)$, and the searched unknown HTC in red $\left(-k\frac{\partial T}{\partial n} = HTC(T - T_\infty)\right)$.

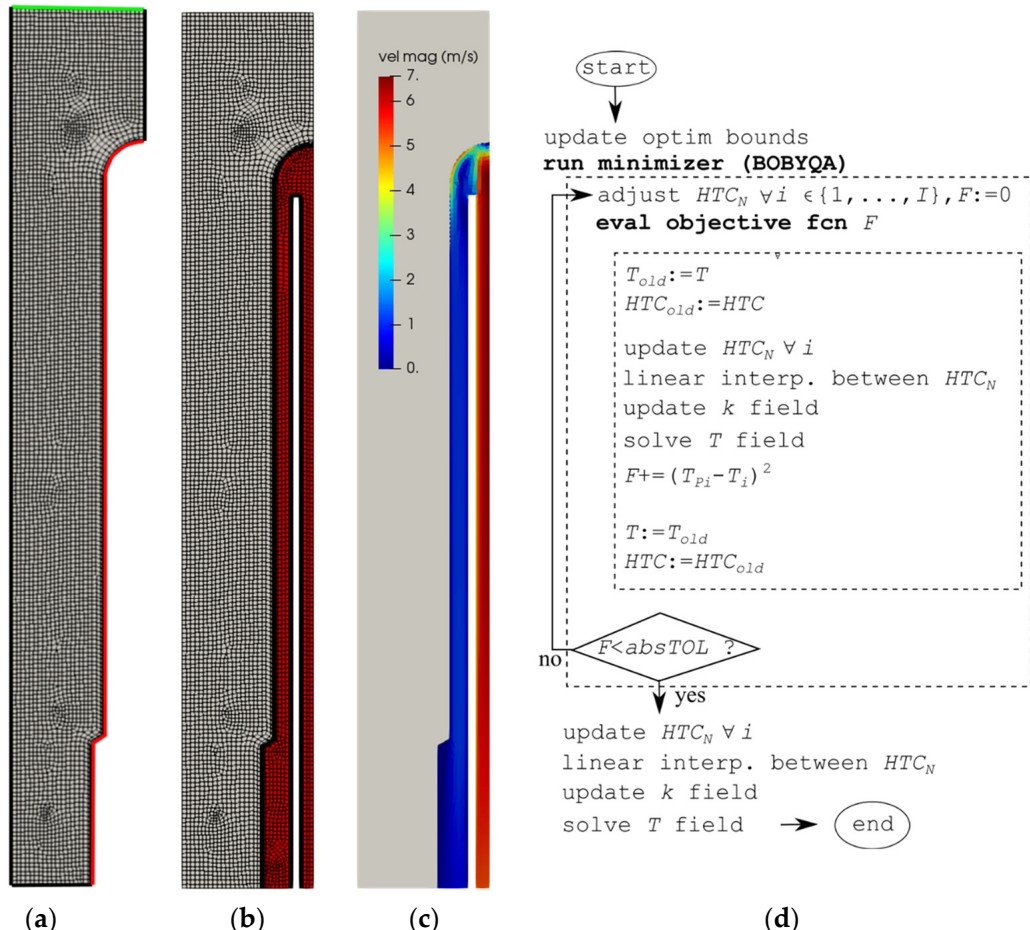

**Figure 3.** The computational domain geometry and mesh (5000 cells) used (**a**) in the IHCP solution and (**b**) in the numerical experiment; (**c**) corresponding velocity field in the numerical experiment; (**d**) the algorithm of the IHCP solver as used in the present study.

The heat flux $q$ (BC in green) is constant in time and space, easily calculated from the cooling power (Table 1). The reference temperature $T_\infty$ is calculated as an arithmetic average of the inlet and outlet temperature of the cooling water (Table 1). The unknown heat transfer coefficient $HTC$ is assumed to vary stepwise linearly along the curved surface (shown as grey trapezoids in Figure 1b). The stepwise linear interpolation is carried out between $I$ discrete nodes $Ni$ (shown as cyan markers in Figure 1b) that are constructed

using an orthogonal projection of the thermocouple tips (shown as red markers in Figure 1b) to the curved surface of the sample.

The initial temperature of the sample $T_0$ was set constant to 300 °C, although any other value would also work due to the fact that Equation (1) does not contain a transient term.

The optimization algorithm (BOBYQA—Bounded Optimization By Quadratic Approximation [29]) is used to find proper values of heat transfer coefficients $HTC_{Ni}$ in each discrete node $Ni \in \{1, \ldots, I\}$ by solving the least-square problem (Equation (2)):

$$\forall \text{ thermocouple } i \in \{1, \ldots, I\} \text{ find } HTC_{Ni} \text{ so that } F = min \sum_{i=1}^{I} (T_{Pi} - T_i)^2, \qquad (2)$$

in which $T_{Pi}$ and $T_i$ are, respectively, the temperature measured using the $i$th thermocouple (Table 1) and the simulated temperature at the location of the $i$th thermocouple tip. A diagram of the IHCP solver is shown in Figure 3d. The parameters of the IHCP solver are summarized in Table 2. A maximum wall-clock time (maxTime) was considered as the stopping criterion of the BOBYQA algorithm.

**Table 2.** Parameters of the IHCP calculation.

| Parameter Name | Value |
| --- | --- |
| ddtScheme | steadyState |
| grad (T) | Gauss linear |
| Laplacian (k,T) | Gauss linear Corrected |
| interpolation | linear |
| snGradSchemes | orthogonal |
| under-relaxation | none |
| linear solver | PCG FDIC (tolerance $1 \times 10^{-14}$) |
| nNonOrthogonalCorrectors | 1 |
| minimizer | BOBYQA (bounded quadratic) |
| maxTime (s) | 300 |
| initial HTC ($Wm^{-2}K^{-1}$) | 10,000 |
| lower bound of HTC ($Wm^{-2}K^{-1}$) | 0 |
| upper bound of HTC ($Wm^{-2}K^{-1}$) | 100,000 |

*2.3. Verification of the IHCP Solver Using a Numerical Experiment*

In the first step, a steady-state fluid flow simulation (Figure 3b,c) with a conjugated heat transfer is carried out in ANSYS FLUENT 17.2, representing a numerical experiment. The temperatures are recorded at the locations of the thermocouple tips, as conducted in the real experiment. Unlike it, the numerical experiment has many advantages over the real experiment, including the following:

- There is no error regarding the temperature measurements (the exact values are defined at the exact locations);
- There is no error in defining the boundary conditions;
- There is no error in selecting the thermophysical properties;
- There is no error in specifying discretization and solution.

Note that "no error" only holds between the numerical experiment and subsequent IHCP calculation and is hence relative. The numerical experiment by no means replaces the above-mentioned experiment. Here, it is exclusively purposed to verify the IHCP solver. Nevertheless, the set-up of the fluid flow model resembles Exp. #2 from Table 1 using the same flow rate, the inlet temperature of the coolant (water), and the cooling power. The fluid flow is modeled assuming the $k$-$\varepsilon$ realizable turbulence RANS model. The thermophysical properties agree with those of the sample used in the experiment (Dievar). A temperature-dependent thermal conductivity was considered, namely, $k = \{30, 31, 32\}$ $Wm^{-1}K^{-1}$ at $T = \{1, 400, 600\}$ °C.

After the fluid flow simulation, the IHCP solver (Figure 3a,d) is fed with the temperature data from the numerical experiment, and a proper spatial distribution of the heat transfer coefficient is determined, as shown in Figure 4a. The surface temperature and the heat flux density are shown in Figure 4b and Figure 4c, respectively. Keeping in mind that the heat flux density is eventually the boundary condition of interest, a very good agreement is observed between the numerical experiment and the IHCP calculation, as seen in Figure 4c. Obviously, the only discrepancies appear due to the stepwise linear interpolation of the HTC. The IHCP calculation converges nearly ideally towards the numerical simulation ($F \approx 0$), which is clearly demonstrated by showing the temperatures at the thermocouple tips side by side in Table 3.

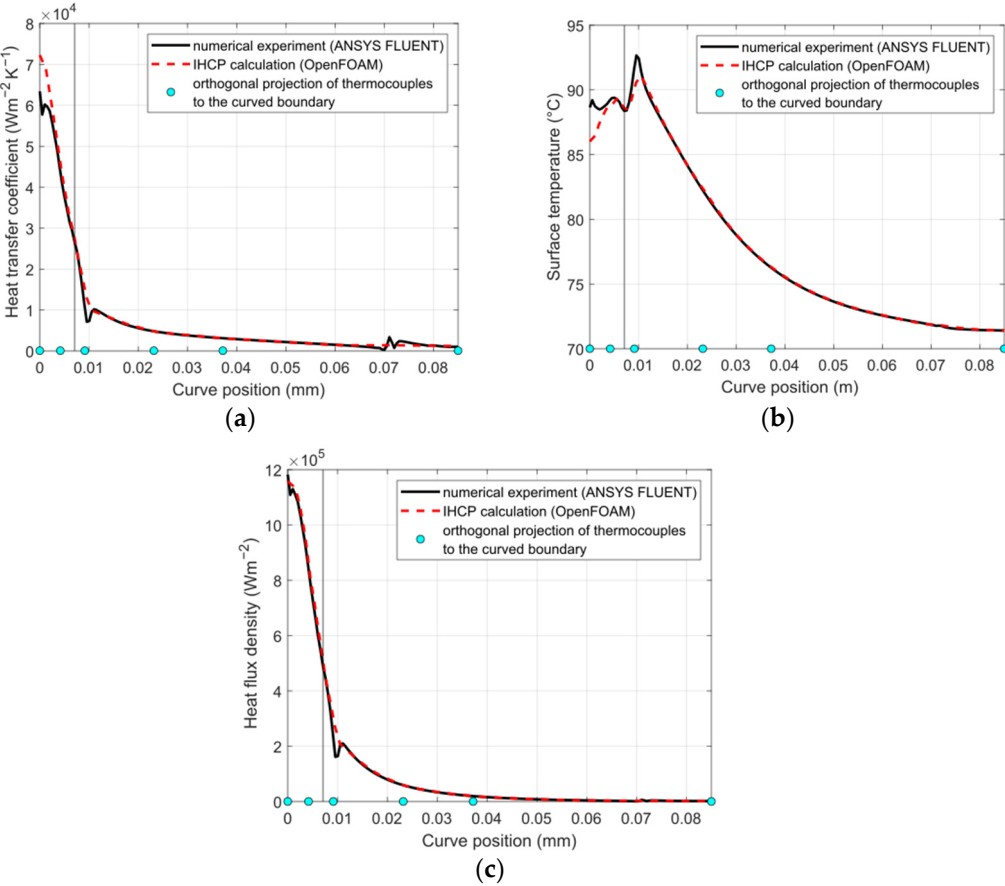

**Figure 4.** Distribution of (**a**) HTC, (**b**) surface temperatures, and (**c**) heat flux density as a function of the curve length for the numerical experiment (solid black lines) and IHCP (dashed red curves). A vertical black line marks the transition from the hemispherical tip to the straight annular section.

**Table 3.** Numerical experiment vs. IHCP calculation: temperatures at the thermocouple tips (the red circle markers in Figure 1b).

| Thermocouples: | #1 | #2 | #3 | #4 | #5 | #6 |
|---|---|---|---|---|---|---|
| Temperatures in num. experiment (°C) | 190.447 | 143.354 | 113.892 | 87.237 | 77.483 | 71.516 |
| IHCP calculation (°C) | 190.447 | 143.353 | 113.891 | 87.237 | 77.483 | 71.516 |

Note that the HTC profile strongly resembles the well-known shape of an impinging jet, as already shown in [7]. Also, note that it is bound to a convective single-phase heat transfer, which might not be the case for the real die insert due to the occurrence of boiling phenomena, particularly in the vicinity of the hemispherical tip [11]. Moreover, the two-equation $k$-$\varepsilon$ realizable turbulence model was used, which is known to underpredict

the development of the turbulence in the wall jet region, thus failing to reproduce the secondary HTC peak therein. Thereby, further physical experiments and subsequent IHCP calculations are mandatory for this study and are discussed in the next section.

### 2.4. IHCP Calculations with Experimental Data

The experimental temperatures and the cooling power from Table 1 are used as input for the IHCP calculations, each of those taking around 5 min of CPU time. High-performance parallel computing is not exploited in this study due to the small axisymmetric numerical grid with 5000 cells (Figure 3a). Roughly 800 BOBYQA iterations are needed to reach the converged solutions. An individual inverse task is terminated when the local change of HTC goes below 1 Wm$^{-2}$K$^{-1}$. Similar to the previous section with the numerical experiment, the IHCP calculation perfectly converges at the same rate towards the experimental data, i.e., reaching $F \approx 0$ (Table 4). In addition, the temperature difference between the measured and the simulated temperature is provided for each thermocouple #1–#6 and each experiment in Table 4. More details about solver and discretization settings, as well as the parameters of BOBYQA minimizer, can be found in [26]. The measured and calculated uncertainties agree very well with the analysis discussed in [30].

**Table 4.** Convergence of the IHCP calculations with the experimental data as an input.

|  |  | Exp. #1 | Exp. #2 | Exp. #3 | Exp. #4 |
|---|---|---|---|---|---|
| Goal function $F$ (K$^2$) |  | 0.242 | 0.135 | 0.0109 | $1.22 \times 10^{-7}$ |
| $T_{Pi} - T_i$ | #1 | −0.093 | −0.099 | −0.00034 | −0.000033 |
| for thermocouples (K): | #2 | 0.202 | 0.215 | 0.00115 | 0.000062 |
|  | #3 | −0.144 | −0.146 | −0.00261 | −0.000128 |
|  | #4 | −0.112 | 0.055 | 0.01609 | 0.000026 |
|  | #5 | −0.247 | 0.115 | −0.05003 | 0.000007 |
|  | #6 | 0.314 | 0.203 | 0.09037 | −0.000318 |

To keep consistency for the representation of the reverse task calculations and their comparison with the physical experiment, the obtained results are displayed as a spatial distribution of the heat transfer coefficient (Figure 5a), surface temperatures (Figure 5b), and heat flux density (Figure 5c), as carried out in the previous section with the numerical experiment used as a reference. A corresponding discussion follows up in the next section.

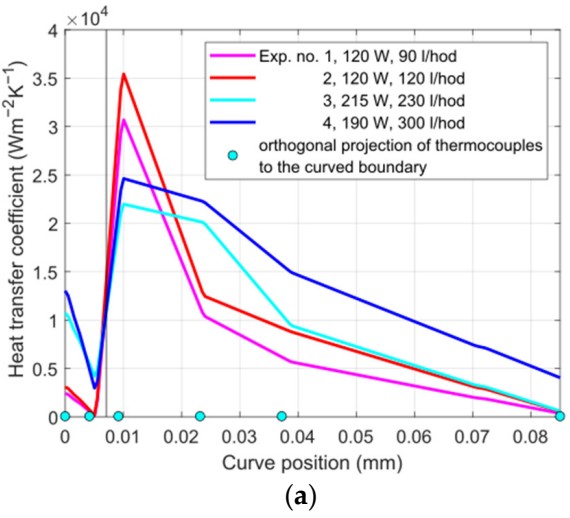

(a)

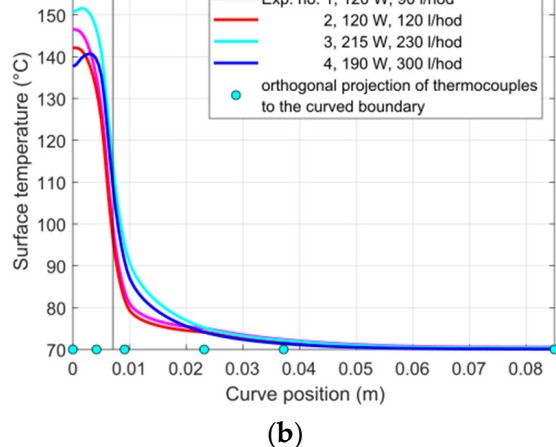

(b)

**Figure 5.** *Cont*.

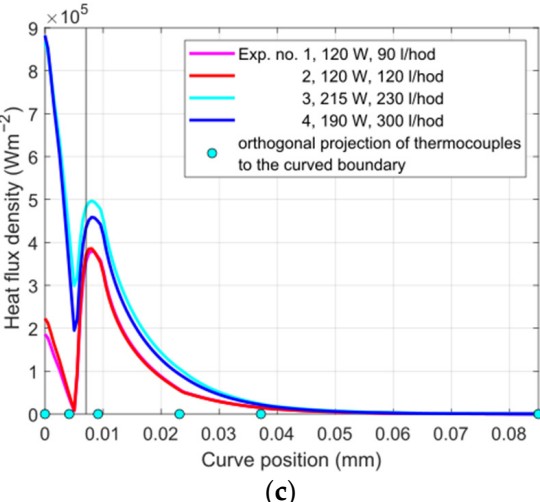

(**c**)

**Figure 5.** Distribution of (**a**) HTC, (**b**) surface temperatures, and (**c**) heat flux density as a function of the curve length obtained from the IHCP calculations using the experimental data from Table 1. A vertical black line marks the transition from the hemispherical tip to the straight annular section.

## 3. Results

An expectedly significant difference was observed between the numerical (Figure 4) and physical experiments (Figure 5). The results in the first (numerical) study are quite similar to the well-known case of a jet impinging upon a hot flat plate, without a significant secondary HTC peak in the wall jet region. A peak of the maximum heat flux density co-locates with the footprint of the jet, followed by a gradual decay as the radial coordinate grows. Similar profiles of the heat flux density and the heat transfer coefficient can be found in [31–33].

The physical experiment is more complex since it involves (i) the effect of the curved surface (hemispherical tip), (ii) the secondary HTC peak in the wall jet region [34], (iii) jet impingement in a confined space resulting in a shift of the maximum heat flux from the stagnation point radially outwards [35], (iv) the occurrence of boiling that is justified by the surface temperatures well above the saturation temperature within the entire region of the hemispherical tip, as shown in Figure 5b.

Despite having only three thermocouples around the hemispherical tip, the behavior of the heat flux density is quite logical. At the stagnation point, i.e., on the axis of symmetry, the relatively high value of heat flux is experienced due to either the undeveloped thermal boundary layer or the onset of the nucleate pool boiling. Soon after that, the velocity boundary layer gradually develops, which is also followed by the development of the thermal boundary layer, which ultimately results in a significant decrease in the local heat flux [36]. Then, the violent transition to turbulence in the wall jet region (Figure 3c) is responsible for the secondary HTC peak. Finally, the heat flux density gradually decreases along the length of the annular section and asymptotically approaches the aforementioned correlation [8].

In Figure 6, the HTC curves from the presented study are shown (in grey) in comparison to the values that would be used in the casting simulation software Magmasoft [11]: one is used to predict a single HTC value in the hemispherical tip, and the other is used to predict a single HTC value in a segment of the annular section. In the remaining part of the annular section, a zero HTC value is considered. The present HTC results qualitatively agree with these correlations: (i) low HTC values in the hemispherical tip and (ii) relatively high values in the segment of the annular section. The well-known HTC correlations are also shown for the laminar and turbulent jet impinging onto a hot flat plate [36,37]. While the presented results resemble the HTC trend in the annular section, they significantly deviate in the hemispherical tip. This discrepancy is explainable since the correlations in references [36,37] consider the free-surface jets only and, thus, do not take into account the

flooding of the hemispherical tip, which leads to the possible entrapment of the air bubbles or vapor pockets.

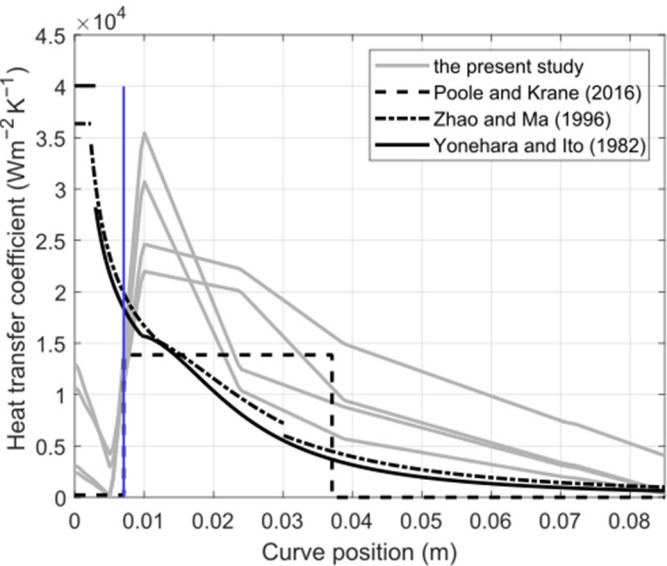

**Figure 6.** Heat transfer coefficient in the presented study vs. correlations from literature [11,36,37]. A vertical blue line marks the transition from the hemispherical tip to the straight annular section.

## 4. Discussion

The present IHCP solver is not limited by any maximum number of thermocouples. Therefore, the resolution of the HTC distribution can be easily increased by considering more thermocouples in the sample. It might be particularly useful in places at which steep changes in heat transfer can be anticipated, e.g., the section with the hemispherical tip. A more detailed or finer HTC distribution can help to further improve the accuracy of casting simulations frequently performed by foundries' R&Ds. If a manufacturer of jet coolers had a specific requirement for the spatial distribution of HTC, e.g., to maximize it or to make it more uniform, the IHCP solver could be used in the design of these products. As a semi-experimental tool, it is suitable for the verification of fluid flow simulations [38], commonly used for the same purpose as the jet cooler design.

Despite being developed and tested with two-dimensional axisymmetric geometry, the present model can be relatively easily extended to 3D with the main challenge being a more complex interpolation of heat transfer coefficients on a three-dimensional surface and an inevitable need to parallelize the optimization subroutines involved in the code.

The IHCP solver is currently limited to steady-state heat conduction problems. However, the authors plan to modify the solver soon to make it transient, in accordance with their 1D IHCP solver published earlier [26]. The authors view the position of thermocouples and the thermophysical properties of the sample as the main source of uncertainties. The authors identify drilling the thermocouple holes as the main difficulty in the preparation of the sample. It is expected that their next research will consider thermocouples with an outer diameter ranging between 0.25 mm and 0.5 mm. Furthermore, the distance of the thermocouple tip from the surface will need to drop below 1 mm; hence, computed tomography will assist when drilling the holes into the required depths as well as later in the determination of the exact locations of the thermocouple tips.

## 5. Conclusions

A solver for the inverse heat conduction problem (IHCP) was developed and verified using the numerical experiment of a die insert. The solver can be used for any axisymmetric or general two-dimensional geometries. It takes an arbitrary number of thermocouples distributed around a curved surface, for which a spatial distribution of heat transfer coefficient

is to be determined. The IHCP solver is based on the open-source code formerly published by the authors and is available for free use at the public repository of reference [26].

The IHCP solver was used to analyze the cooling intensity of a realistic die insert using data from temperature measurements. It was shown that the spatial heat flux density strongly deviates from the single-phase CFD calculations and from the distribution anticipated from the canonical relations for the jet impinging onto a hot surface. The presented approach and results provide the methodology and tools to reconstruct the correct thermal boundary condition for an arbitrary die insert and can be, thus, used in a design process.

The authors remain active in this important research topic. Similar measurements for other cooling elements are being performed in cooperation with the company ESI Group, which develops the ProCAST program for complex foundry analyses, including HPDC technology. Future studies are focused on IHCP calculations using the presented method and new data obtained from a unique experiment with casting an aluminum melt into a mold sample, which is cooled using a die insert equipped with many thermocouples.

**Author Contributions:** Conceptualization, methodology, investigation, J.B. and K.M.; software, J.B. and A.V.; validation, resources, V.K. (Vladimir Krutis) and V.K. (Vaclav Kana); writing—original draft preparation, J.B. and A.V.; writing—review and editing, K.M., V.K. (Vladimir Krutis), V.K. (Vaclav Kana), E.K.-S. and A.K.; supervision, project administration, funding acquisition, V.K. (Vladimir Krutis) and A.K. All authors have read and agreed to the published version of the manuscript.

**Funding:** The project TN02000010/03 of the National Competence Centre of Mechatronics and Smart Technologies for Mechanical Engineering is co-financed from the state budget by the Technology Agency of the Czech Republic within the National Centres of Competence Programme. The authors acknowledge the financial support by the Austrian Federal Ministry of Economy, Family and Youth and the National Foundation for Research, Technology and Development within the framework of the Christian Doppler Laboratory for Metallurgical Applications of Magnetohydrodynamics.

**Data Availability Statement:** Data are contained within this article.

**Acknowledgments:** Computational resources were supplied by the project "e-Infrastruktura CZ" (e-INFRA CZ LM2018140) and supported by the Ministry of Education, Youth and Sports of the Czech Republic. Computational resources were provided by the ELIXIR-CZ project (LM2018131), part of the international ELIXIR infrastructure.

**Conflicts of Interest:** The authors declare no conflict of interest.

## Nomenclature

| | |
|---|---|
| $F$ | objective or target function ($K^2$) |
| $HTC$ | heat transfer coefficient ($Wm^{-2}K^{-1}$) |
| $k$ | thermal conductivity ($Wm^{-1}K^{-1}$) |
| $q$ | heat flux ($Wm^{-2}$) |
| $T$ | temperature (K) |
| $\Omega$ | space domain (m) |
| Abbreviations | |
| BOBYQA | Bounded optimization by quadratic approximation |
| BC | Boundary condition |
| CFD | Computational fluid dynamics |
| Dievar | High-performance chromium–molybdenum–vanadium-alloyed hot work tool steel |
| HPDC | High-pressure die casting |
| NLopt | Nonlinear optimization toolbox |
| OpenFOAM | Open-source CFD package |
| RANS | Reynolds-averaged Navier–Stokes (equations) |
| PDE | Partial differential equation |
| PID | Proportional–integral–derivative controller |



Subscripts

| | |
|---|---|
| $i$ | summation index |
| $I$ | number of thermocouples |
| $N$ | a point on the curved boundary of the jet cooler generated by the orthogonal projection of the thermocouple tip, i.e., the point $P$ |
| $P$ | measuring point of thermocouple |

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
