# Peer review of "Experimental and Numerical Investigations into Heat Transfer Using a Jet Cooler in High-Pressure Die Casting"

_jmmp, doi:10.3390/jmmp7060212_

Round 1

Reviewer 1 Report

Comments and Suggestions for Authors

The presented study deals with the estimation of boundary conditions inside a bubbler used in high-pressure die-casting dies. The topic is relevant and within the scope of the Journal of Manufacturing and Materials Processing.

I must admit that I have enjoyed reading this manuscript and find the study scientifically sound and clearly presented. Many recent papers severely lack proper verification of the numerical model, not this one. The paper should be published. However, I believe that the paper can, and should, be improved prior to publishing by incorporating the following suggestions:

  1. Page 1, line 38: "...is a fancy topic of many publications [5]" I think that there is no need for implied dismissal of conformal cooling. This technology indeed offers advantages and I have personally seen it being increasingly implemented in real-world applications outside pure research.
  2. It would be beneficial to give some reasoning for using selected values of coolant flow rates and die temperatures. For example, die inserts in HPDC tools are exposed to both higher and lower temperatures (depending on geometry, and the exact moment within a casting cycle). Therefore, investigating the full parameter envelope would be beneficial. 
  3. Page 2, second paragraph and Table 1 should be deleted. I think that counting the number of papers returned by some databases for specific keywords is not worthy of archival publication. If nothing else, my search in Scopus for "bubbler" AND "die casting" returned only 2 results [7] and [10] - both relevant. Such information is suitable for large Meta-analysis studies, that were not done here. 
  4. More details on experimental and numerical setup should be given. Mainly, the manuscript only mentioned using Deviar (Uddeholm?) as a die material. Used thermophysical properties should be included since they influence the accuracy of the inverse model. Additionally, some notes on the surface roughness of the bore surface should be included. Ideally, profilometry results should be included. However, I am aware that they are probably not available. So, at least a general remark on surface condition as finely polished, as-drilled, bored,... should be given.
  5. The manuscript mentions that the goal function F reached near zero values. However, as defined in eq. 2, that can mean a lot of things: where some temperatures could be significantly overestimated, while others were underestimated. The residuals for each thermocouple should be given. 
  6. Figure 6 mentions Fu [9], please check that there was no mixup between references 9, 11, or some other reference. I think I did not see any of them give such HTC curve.  Other mentioned references are also in question.
  7. If possible, as a time-tested open-source practice, I would suggest including the developed OpenFOAM solver as supplementary material for future use. But, I would understand if authors plan to refrain from this.

Author Response

Dear Reviewer, please see the attachment. Author

Reviewer 2 Report

Comments and Suggestions for Authors

Dear authors, thanks for sharing your work with the rest of the research  community.

Summary: 

The paper deals with an experimental approach to tune the CFD simulations of heat extraction from mold cavity in aluminium HPDC.

General comments:

The approach resembles others from analogue industrial problems, that is, starting with a very standardized and well characterized geometry which allows to develop simulatoin models that can be extrapolated to more complex systems.

The manufacturing challenge that is addressed is relevatn for the HPDC field as it directly impacts tack time (and thus productivity) and can be used to improve homogeneous cooling parts with severe section thickness changes (strong thermal moduli gradients). Furthermore, HPDC of Gigacastings is a trending topic in automotive industry and the work is a basis for future development in this subject.

The experimental setup is properly described and reproducible by third party research teams.

Models are in good agreement with the experimental results.

The step towards more complex geometries could be better explained in the Results section (as there is not a Discussion section).

Specific comments:

Taking into acocunt the content and the extension, I suggest the authors to consider changing the Article Type to "Case report" or "Technical note" since the work is somwhere between an engineering work and research and development activity.

Author Response

Dear Reviewer, please see the attachment. Author.

Reviewer 3 Report

Comments and Suggestions for Authors

This article conducts in-depth research on the heat transfer of jet coolers in high-pressure die-casting. The use of numerical simulation and experimental temperature measurement increases the credibility of research results. The author effectively summarized the relevant literature on jet cooling and heat transfer, indicating their understanding of the research background and significance. The results of temperature measurement and heat transfer calculation are clearly presented and support these conclusions. Modification suggestions: 

1) More information on the importance of high-pressure die-casting and the role of heat transfer in casting and mold system design will improve the clarity of research objectives, which can be further reflected in the introduction. 

2) In the methodology section, it would be beneficial to provide more details on the specific numerical simulation settings used in the OpenFOAM software package and the specific parameters used in the solver. This will help readers better understand the method and potentially replicate the simulation. 

3) The section on the solver for inverse heat conduction problems can be expanded to provide a more detailed explanation of mathematical methods. Providing the equations or algorithms used in the solver, as well as any assumptions or limitations, will enhance the technical rigor of the research. 

4) In the conclusion section, how can the obtained heat transfer coefficient distribution be applied to improve the design and performance of high-pressure die-casting jet coolers? It is recommended to include this discussion in the results analysis, as it will increase the relevance of the research results. 

5) This study can analyze and discuss potential limitations and future research directions, explore uncertainties or assumptions in numerical simulation, and provide reference and reference for simulation research in this field.

Comments on the Quality of English Language

Minor editing of English is required.

Author Response

Dear Reviewer, please see the attachment. Authors.

Round 2

Reviewer 1 Report

Comments and Suggestions for Authors

Dear authors,
Please check the abscissa (x-axis) units in Figs. 4-6. I think that there could be a mixup between mm and m.

Author Response

Dear Reviewer,

thank you so much for finding such blatant mistake in the manuscript. We changed mm to m in figs. 4-6. We wish you all the best and once again thank you! Sincerely, JB and AV